# Aberrant cohesin function in *Saccharomyces cerevisiae* activates Mcd1 degradation to promote cell lethality

**Gurvir Singh**, **Robert V. Skibbens**\*

Department of Biological Sciences, Lehigh University, Bethlehem, Pennsylvania, United States of America.

\* rvs3@lehigh.edu

## Abstract

The cohesin complex is composed of core ring proteins (Smc1, Smc3 and Mcd1) and associated factors (Pds5, Scc3, and Rad61) that bind via Mcd1. Extrusion (looping from within a single DNA molecule) and cohesion (the tethering together of two different DNA molecules) underlie the many roles that cohesins play in chromosome segregation, gene transcription, DNA repair, chromosome condensation, replication fork progression, and genome organization. While cohesin functions flank the activities of critical cell checkpoints (including spindle assembly and DNA damage checkpoints), the extent to which checkpoints directly target cohesins, in response to aberrant cohesin function, remains unknown. Based on prior evidence that cells mutated for cohesin contain reduced Mcd1 protein, we tested whether loss of Mcd1 is based simply on cohesin instability or integrity. The results show that Mcd1 loss persists even in *rad61* cells, which contain elevated levels of stable chromosome-bound cohesins, and also in *scc2–4*, which do not affect cohesin complex integrity. In fact, re-elevating Mcd1 levels suppresses the temperature-sensitive growth defects of all cohesin alleles tested, revealing that Mcd1 loss is a fundamental mechanism through which cohesins are inactivated to promote cell lethality. Our findings further reveal that cells that exhibit aberrant cohesin function employ E3 ligases (such as San1) to target Mcd1 for degradation. This mechanism of degradation appears unique in that Mcd1 is reduced during S phase, when Mcd1 levels typically peak and despite a dramatic upregulation in *MCD1* transcription. We infer from these latter findings that cells contain a negative feedback mechanism used to maintain Mcd1 homeostasis.

## Author summary

Cohesins are central to almost all aspects of DNA regulation (chromosome segregation, gene transcription, DNA repair, chromosome condensation, replication fork progression, and genomic organization). Cohesin also play key roles in cell

**Data availability statement:** All data are included in the body of the manuscript, in the supplemental information, or the following URL: https://bio.cas.lehigh.edu/faculty-staff/bob-skibbens#scholarship.

**Funding:** This work was suppported by the National Institutes of Health (R15GM139097 to RVS). The funders had no role in the study design, data collection and analysis, decision to publish, or prepration of the manuscript. Salaries/stipends were provided to R. Skibbens and G. Singh by the National Institutes of Health (R15GM139097 to RVS).

**Competing interests:** The authors have declared that no competing interests exist.

checkpoints: cohesin mutations activate the spindle assembly checkpoint while double strand DNA breaks can elicit a new round of cohesin establishment. In the current study, we provide evidence for a novel cohesin surveillance system that employs E3 ligases that directly target Mcd1, a core component of the cohesin ring structure, for degradation. We further describe a feedback mechanism through which cells dramatically induce *MCD1* transcription to maintain Mcd1 homeostasis. Finally, we provide evidence that requires the re-evaluation of phenotypes associated with other cohesin gene mutations.

## Introduction

The identification of checkpoints has produced significant clinical advances over the last half century [1,2]. After the discovery that the tumor suppressor *TP53* gene is mutated in more than 50% of all cancers, advances from both basic science and clinical settings led to new strategies through which p53 can be re-activated, mutated p53 degraded, or synthetic lethal mechanisms to eliminate cancer cells [3–7]. Similarly, highly proliferative cancer cells are disproportionally sensitive to spindle assembly checkpoint inhibitors, compared to non-tumorigenic cells [8–10]. Often, factors that exhibit multiple functions, or reside at the nexus of critical pathways, are monitored by surveillance mechanisms and provide important avenues to improve human health.

Cohesins are ATPase protein complexes composed of a core ring (Smc1, Smc3 and Mcd1/Scc1, herein Mcd1) and associated factors (Rad61, Pds5 and Scc3/Irr1, herein Scc3) that bind via Mcd1 [11–17]. Cohesins are central to almost all aspects of DNA regulation (chromosome segregation, gene transcription, DNA repair, chromosome condensation, replication fork progression, and genomic organization) [11,12,14,18–32]. Underlying this complex output of roles are two very distinct cohesin functions: extrusion (looping from within a single DNA molecule) and cohesion (the tethering together of two different DNA molecules) [11,12,21,22,25–27]. Mutations in cohesins that affect DNA looping can give rise to severe developmental abnormalities such as Cornelia De Lange Syndrome (CdLS) and Roberts Syndrome (RBS). These multifaceted maladies often include intellectual disabilities, hearing loss, microcephaly, phocomelia and abnormalities in the heart and gastrointestinal tract [33–39]. Mutations that impact tethering give rise to aneuploidy - a hallmark of cancer cells [40–42]. Recent evidence indeed suggests that cancer cells rely on elevated cohesin activity for survival [43–47].

It is well established that cohesin functions flank critical cell checkpoints. For instance, cohesin mutations that abolish sister chromatid cohesion (tethering) activate the spindle assembly checkpoint [48–50], consistent with classic micromanipulation and laser ablation studies that cells monitor chromosome biorientation, and thus tension, produced across the two mitotic half-spindles [51–56]. In contrast, double strand DNA breaks activate ATM/ATR and CHK1 kinases that elicit a *de novo* round of cohesin deposition and cohesion establishment both at sites of damage and

genome-wide [18,24,57–63]. The remarkable involvement of cohesins in cell cycle checkpoints, both to maintain euploidy and promote error-free DNA damage repair, underscores their central role in maintaining genome integrity.

In this study, we present evidence that cohesins are a direct target of a surveillance system that may cull aneuploid or transcriptionally aberrant cells from a normal population. The current study was motivated, in part, by prior observations that cohesin mutations result in a significant reduction of Mcd1 protein [64–67]. Importantly, the significance of Mcd1 reduction, and mechanisms involved, remained undetermined. Here we show that Mcd1 reduction in cohesin mutant cells is not a benign product of cohesin mutations but significantly contributes to their lethality. Importantly, the loss of Mcd1 occurs independent of defects in either cohesin complex stability or integrity. Finally, we show that cohesin dysfunction activates a surveillance system that employs E3 ligases and the proteasome to specifically target Mcd1 for degradation.

## Results

### Cohesin dysfunction activates a novel mechanism to reduce Mdc1

The majority of cohesin-mutated cells tested to date (*smc1–259*, *smc3–42*, *pds5–1*, *pds5Δ elg1Δ*, and *eco1Δ rad61Δ*) contain significantly reduced levels of Mcd1 protein [64–67]. Eco1/Ctf7 (herein Eco1) is an acetyltransferase that targets Smc3 to promote stable cohesin-DNA association [48,66,68–71]. Rad61 dissociates cohesin from DNA such that *rad61Δ* cells retain elevated levels of stably-bound cohesins [15,72–79]. Given the opposing activities of Eco1 (cohesin stabilization, [48,66]) and Rad61 (cohesin dissociation [15]), we hypothesized that *rad61Δ* cells should retain wildtype levels of Mcd1 and suppress the loss of Mcd1 in *eco1* mutated cells. To test these predictions, log phase cultures of wildtype, *rad61Δ*, *eco1–203*, and *eco1Δ rad61Δ* cells were arrested in early S phase (hydroxyurea, HU) prior to shifting to 37°C (Fig 1A). Surprisingly, Western blot quantifications of the resulting extracts revealed that Mcd1 levels are reduced not only in *eco1* temperature-sensitive (ts) strains, but also significantly reduced in *rad61Δ* cells (Fig 1B, 1C). Nor did the deletion of *RAD61* provide any benefit to *eco1Δ* cells with respect to Mcd1 levels (Fig 1B, 1C). These results suggest that Mcd1 is reduced by a mechanism that is independent of cohesin complex stability.

Given the results above, it became important to test whether the loss of Mcd1, in the absence of cohesin function, is unique or whether other cohesin complex components are similarly reduced? To address this, we examined Pds5 protein levels in *eco1Δ rad61Δ* cells - a factor that relies on Mcd1 for cohesin-association [13,14]. As above, log phase cultures of wildtype and *eco1Δ rad61Δ* cells were arrested in early S phase (HU) prior to shifting to 37°C. Extracts of the resulting cells were then assessed by Western blot for Pds5 levels. The results indicate that, even though Mcd1 protein is indeed reduced, there is no statistical significance between the level of Pds5 protein across wildtype and *eco1Δ rad61Δ* cells (S1 A,B,C Fig). Thus, Mcd1 reduction appears to be a specific feature of cells deficient for cohesin function.

The apparent unique reduction of Mcd1 in *eco1Δ rad61Δ*, *rad61Δ*, and *eco1–1* cells, all of which retain structurally intact cohesin complexes, is remarkable. It thus became important to further test the extent to which Mcd1 levels might be reduced in cells deficient in Scc2 function. Scc2 is essential for cohesin loading but, as neither a core nor auxiliary component, plays no role in cohesin complex assembly or integrity [80–83]. Log phase cultures of wild-type and *scc2–4* cells were arrested in S phase (HU) for 2 hours at 23°C (permissive temperature) and then shifted to 37°C for 1 hour (Fig 2A). Extracts of the resulting cultures were assessed by Western blot to quantify Mcd1 protein levels. The results reveal that Mcd1 is significantly reduced in *scc2–4* mutant cells, compared to wild-type cells (Fig 2B, 2C). We infer from these findings that a novel surveillance system exists that monitors for aberrant cohesin function, rather than cohesin structure, to regulate Mcd1 protein levels.

### A negative feedback loop regulates *MCD1* expression

The near ubiquitous reduction of Mcd1 in cohesin-mutated cells prompted us to uncover the underlying molecular mechanism. Mcd1 is unique among cohesin subunits in that it is the only core subunit that is degraded at anaphase onset (to

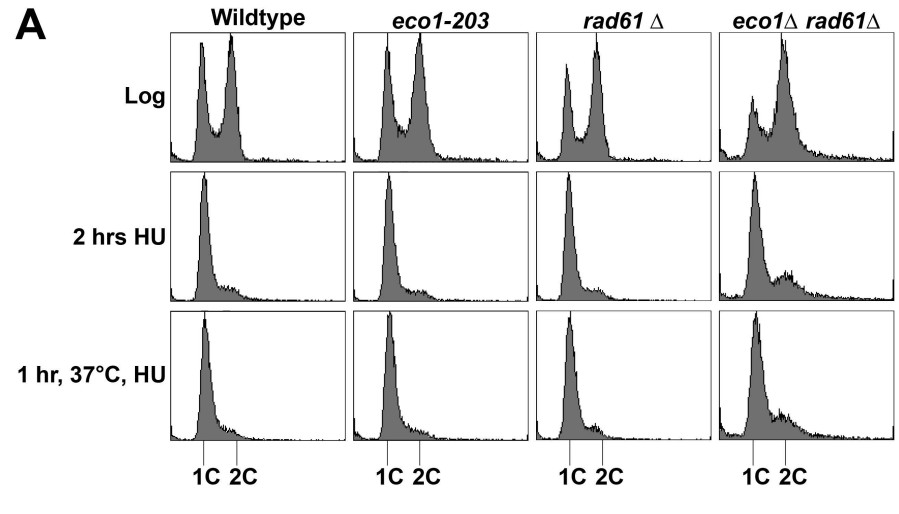

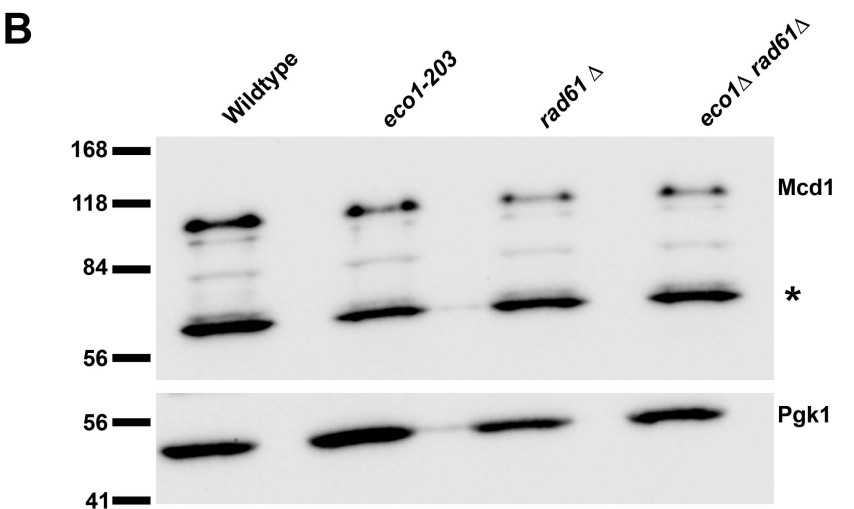

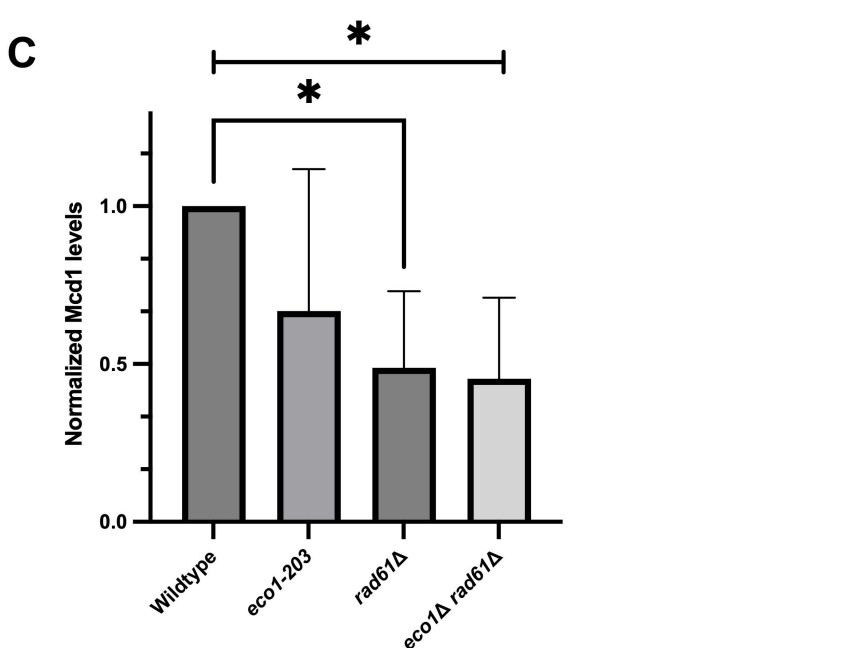

PLOS Genetics

**Fig 1. Mcd1 protein levels are reduced in *eco1-203* and *rad61Δ* mutated cells. (A)** Flow cytometry data of DNA contents for wildtype (YPH499), *eco1-203* (YBS514), *rad61Δ* (YMM808) and *eco1Δ rad61Δ* (YBS829) mutant cells. Log phase cultures were synchronized in S phase at their respective permissive temperatures, 23°C for *eco1-203* and 30°C for wildtype *eco1Δ rad61Δ* and *rad61Δ* mutant cells, then shifted to 37°C for 1 hr. **(B)** Representative Western Blot of Mcd1 (top panel) and Pgk1 (lower panel) protein obtained from extracts of HU-synchronized wildtype, *eco1-203*, *rad61Δ* and *eco1Δ rad61Δ* mutant cells indicated in **(A)**. * indicates non-specific band. **(C)** Quantification of Mcd1, normalized to Pgk1 loading controls. Statistical analysis was performed using a two-tailed *t*-test. Statistical differences (*) are based on a $P < 0.05$ obtained across three experiments (n = 3). Error bars indicate the standard deviation.

allow for sister chromatid segregation), and then transcribed starting at the G1/S transition, each and every cell cycle [11,66,84]. Previous findings documented a complex transcriptional network that regulates *MCD1* expression [65]. It thus became important to test the extent to which *MCD1* transcription is reduced in the *eco1Δ rad61Δ* cells. Log phase wildtype and *eco1Δ rad61Δ* cells were arrested in early S phase (HU) (Fig 3A) - a point in the cell cycle at which *MCD1* expression and Mcd1 protein levels both peak in wildtype cells, but in which Mcd1 protein levels are significantly reduced in *eco1Δ rad61Δ* cells [11,65]. We confirmed that Mcd1 protein levels were significantly reduced in the same *eco1Δ rad61Δ* cultures (Fig 3B, 3C) that we used to test for changes in *MCD1* transcription. In contrast to the model in which *MCD1* transcription is decreased, quantification of qRT-PCR revealed that *MCD1* transcript levels are instead significantly increased (~5.5 fold) in *eco1Δ rad61Δ* cells compared to wildtype cells (Fig 3D). These results indicate that the loss of Mcd1 protein in *eco1Δ rad61Δ* cells is not dependent on reduced *MCD1* transcription. Moreover, our findings reveal a compensatory feedback mechanism in which cells increase *MCD1* transcription in response to decreased Mcd1 protein levels.

### Identification of an E3 ligase mechanism that promotes Mcd1 degradation

Having excluded a transcription-based mechanism, it became important to test the extent to which the reduction in Mcd1 protein level occurs through degradation. Esp1, a caspase-type protease, cleaves Mcd1 during anaphase onset [65,84–86]. If Esp1 cleaves Mcd1 in response to reduced cohesin function, then Mcd1 levels should increase in cells that harbor *ESP1* mutations. To formally test this hypothesis, *eco1Δ rad61Δ* cells and *esp1–1* cells were mated and the resulting diploids sporulated and dissected. Log phase cultures of the resulting strains were then tested for changes in temperature-sensitive growth defects. Surprisingly, reduced Esp1 activity failed to suppress *eco1Δ rad61Δ* cell ts growth defects and in fact further exacerbated the growth defects (S2 A, B Fig). We further note the 83 kDa and 48 KDa fragments produced by Esp1-dependent degradation of Mcd1 [87,88] are largely absent in our Western blots of cells that harbor cohesin defects (Figs 1B, 2B, 6B). These findings largely negate a scenario in which Esp1 is targeting Mcd1 in response to deficits in cohesin function and further support prior evidence that Esp1 activity is predominantly limited to anaphase onset [84,86].

Protein ubiquitination, through E3 ligases, play key roles in numerous cellular activities that include degradation [89,90]. Thus, we focused on E3 ligases as a mechanism to reduce Mcd1 protein levels, a model supported by evidence that Mcd1 is ubiquitinated [91,92]. The E3 ligases that target Mcd1, however, remain unidentified. Thus, we generated *de novo* a candidate list based on genetic or physical interactions reported in either BioGrid or SGD (*Saccharomyces* Genome Database) across the various cohesins subunits [93–96]. These efforts resulted in a list of six candidates (Bre1, Bul2, Ldb19, Ubr1, Das1, and San1), all of which are encoded by non-essential genes. These E3 ligases respond to various cell challenges such as oxidative stress (Bre1), heat-shock responses (Bul2), starvation (Ldb19), N-end rule degradation (Ubr1), cell cycle progression (Das1), and nuclear aggregates (San1) (SGD). We reasoned that if any one of the E3 ligases are in part responsible for ubiquitinating Mcd1, then their deletion should suppress *eco1Δ rad61Δ* cell ts growth defects. To test this model, each of the E3 ligase genes (*BRE1*, *BUL2*, *LDB19*, *UBR1, DAS1* and *SAN1*) were individually deleted from wildtype and *eco1Δ rad61Δ* cells. Log phase cultures of the resulting transformants were serially diluted onto rich medium plates and incubated at either 30°C or 37°C, temperatures respectively permissive and non-permissive for *eco1Δ rad61Δ* cell growth (Fig 4). Deletion of *BUL2* or *UBR1* had no impact on either wildtype or *eco1Δ rad61Δ* cells

**A**

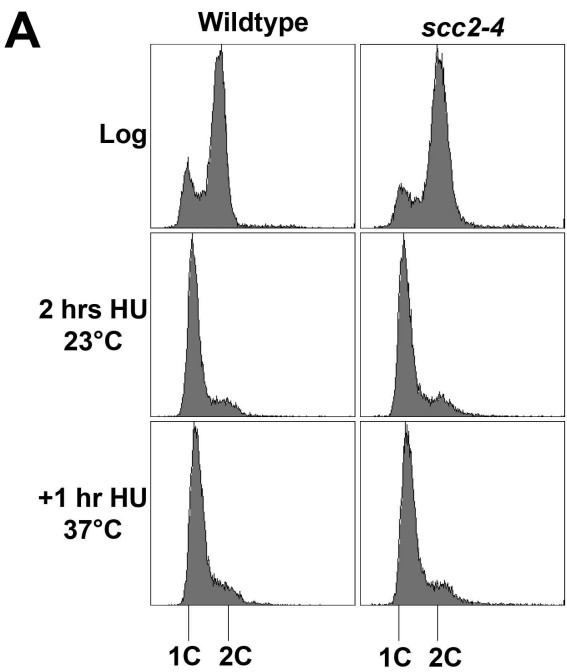

**B**

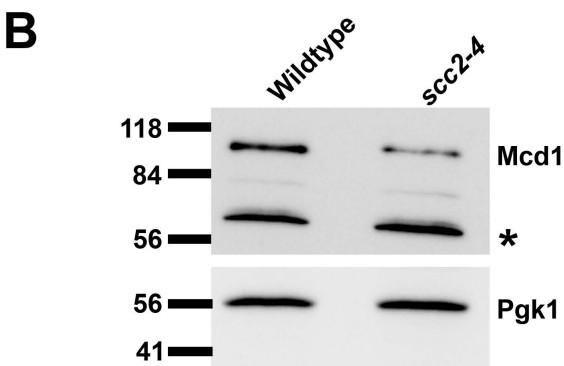

**C**

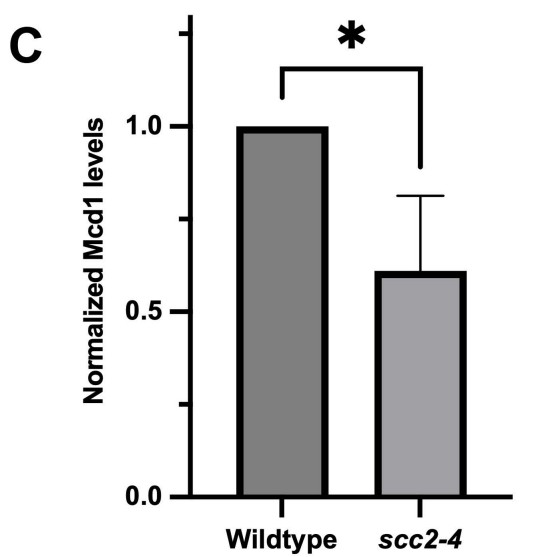

**Fig 2. Mcd1 protein levels are reduced in *scc2-4* mutated cells. (A)** Flow cytometry data of DNA content for log phase of wildtype (YBS2031/KT046) and *scc2-4* cells (YBS2033/KT048) arrested in S phase at 23°C for 2 hours and shifted to 37°C for 1 hour. **(B)** Western Blot of Mcd1 (top panel) and Pgk1 (lower panel) protein obtained from extracts of HU-synchronized cells indicated in **(A)**. * indicates non-specific band. **(C)** Quantification of Mcd1, normalized to Pgk1 loading controls. Statistical analysis was performed using a two-tailed *t*-test. Statistical differences (*) are based on a *P* < 0.05 obtained across three experiments (n = 3). Error bars indicate the standard deviation.

at either temperature (Fig 4A, 4D). Deletion of *BRE1* exhibited an adverse effect on both wildtype and *eco1Δ rad61Δ* cells (Fig 4B), consistent with a prior report that *bre1Δ* cells exhibit genomic instability [97]. Compared to the adverse but non-specific impact of *BRE1* deletion, deletion of *LDB19* produced a severe negative impact specific to *eco1Δ rad61Δ* cells (Fig 4C). In contrast to the results above, deletion of *SAN1*, and to a lesser extent *DAS1*, suppressed the ts growth defects otherwise exhibited by *eco1Δ rad61Δ* cells (Fig 4E, 4F).

The findings above suggest that the E3 ligase San1 plays a critical role in Mcd1 degradation. If correct, then *eco1Δ rad61Δ* cells further deleted of *SAN1* should retain higher levels of Mcd1, compared to *eco1Δ rad61Δ* cells. To directly test this prediction, log phase cultures of *eco1Δ rad61Δ* cells and *eco1Δ rad61Δ san1Δ* cells were arrested in S phase (HU) and the resulting extracts assessed by Western blot (Fig 5A, 5B). The results document that Mcd1 protein levels are significantly increased (~36%, based on 3 biological replicates) in *eco1Δ rad61Δ* cells that are further deleted of *SAN1* (Fig 5C). In summary, our genetic findings that deletion of *SAN1* from *eco1Δ rad61Δ* cells partially suppresses the ts phenotype, combined with biochemical studies that *SAN1* absence leads to an increase of Mcd1 levels, provide strong support for the model that cohesin dysfunction activates E3 ubiquitin ligase-mediated degradation of Mcd1. However, because *SAN1* deletion only partially rescues the *eco1Δ rad61Δ* phenotype, additional E3 ligases involved in Mcd1 turnover are likely involved.

### The proteasome degrades Mcd1 in *eco1 rad61* cells during mitosis

The above findings are noteworthy both for documenting reduced Mcd1 levels during S phase (when Mcd1 protein and *MCD1* expression levels peak during the cell cycle) and for the identification of an E3 ligase pathway that targets Mcd1 for degradation. Together, these results raise the possibility that cells respond to cohesin dysfunction by promoting Mcd1 degradation via the proteosome, potentially in a cell cycle manner. To test this prediction, *PDR5*, which encodes a drug efflux pump, was deleted in *eco1Δ rad61Δ* cells to allow for the retention of MG-132 - a proteasome inhibitor [98]. We first tested whether proteosome inhibition might result in the retention of Mcd1 during S phase. Log phase cultures of *eco1Δ rad61Δ pdr5Δ* triple mutant cells were arrested in S phase for 2 hours, followed by treatment with MG-132 or DMSO (control) in the presence of hydroxyurea (S3 A Fig). Remarkably, changes in Mcd1 protein levels remained similar across cells incubated in either DMSO or MG-132 (S3 B,C Fig), suggesting that Mcd1 degradation occurred elsewhere in the cell cycle. Since Mcd1 is essential during M phase, and minimally produced in G1 [11], we assessed the extent to which Mcd1 degradation occurs prior to anaphase onset. Log phase cultures of *eco1Δ rad61Δ pdr5Δ* triple mutant cells were incubated in media that contains nocodazole (to synchronize cells preanaphase) along with either MG-132 (proteosome inhibitor) or a DMSO control. The progression of cells from log-phase growth to a pre-anaphase arrest was monitored by flow cytometry (Fig 6A). Extracts of the resulting cultures were then assessed by Western blot (Fig 6B). Quantification of the resulting blots reveal that Mcd1 levels are significantly elevated (nearly 3 fold) in cells incubated in the presence of MG-132, compared to the DMSO control (Fig 6C). Together, these results reveal that degradation occurs prior to anapahase onset, when Mcd1 is essential for sister chromatid cohesion, through a post-ubiquitination process performed by the proteosome. The extent to which Mcd1 is similarly degraded in G1 or G2 phases remains to be determined in future studies.

### *MCD1* overexpression rescues the inviabilty of cohesin mutated cells

The reduction in Mcd1 protein levels is an attribute common to all cohesin mutated cells tested to date (64–67, current study). This near ubiquitous reduction prompted us to ask the following question: are ts growth defects due to the mutated

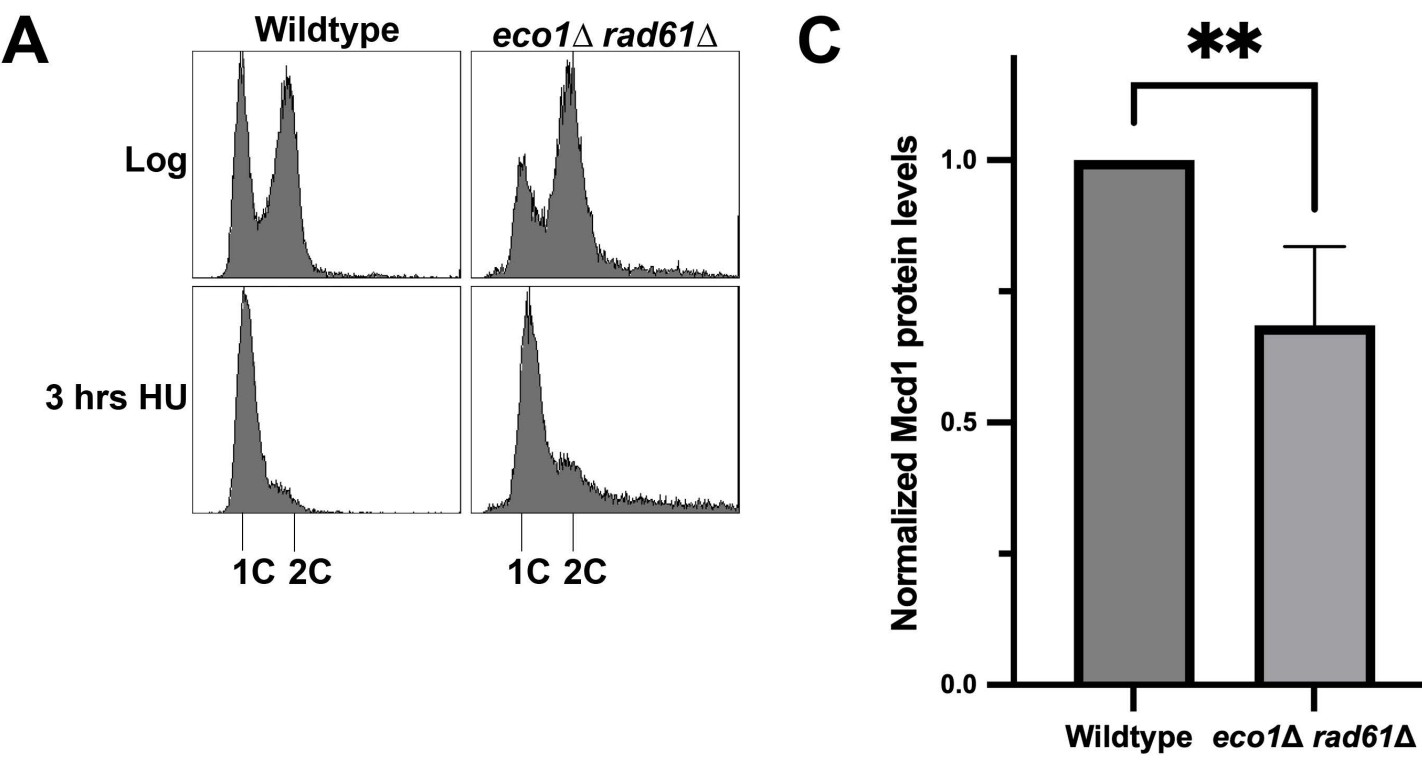

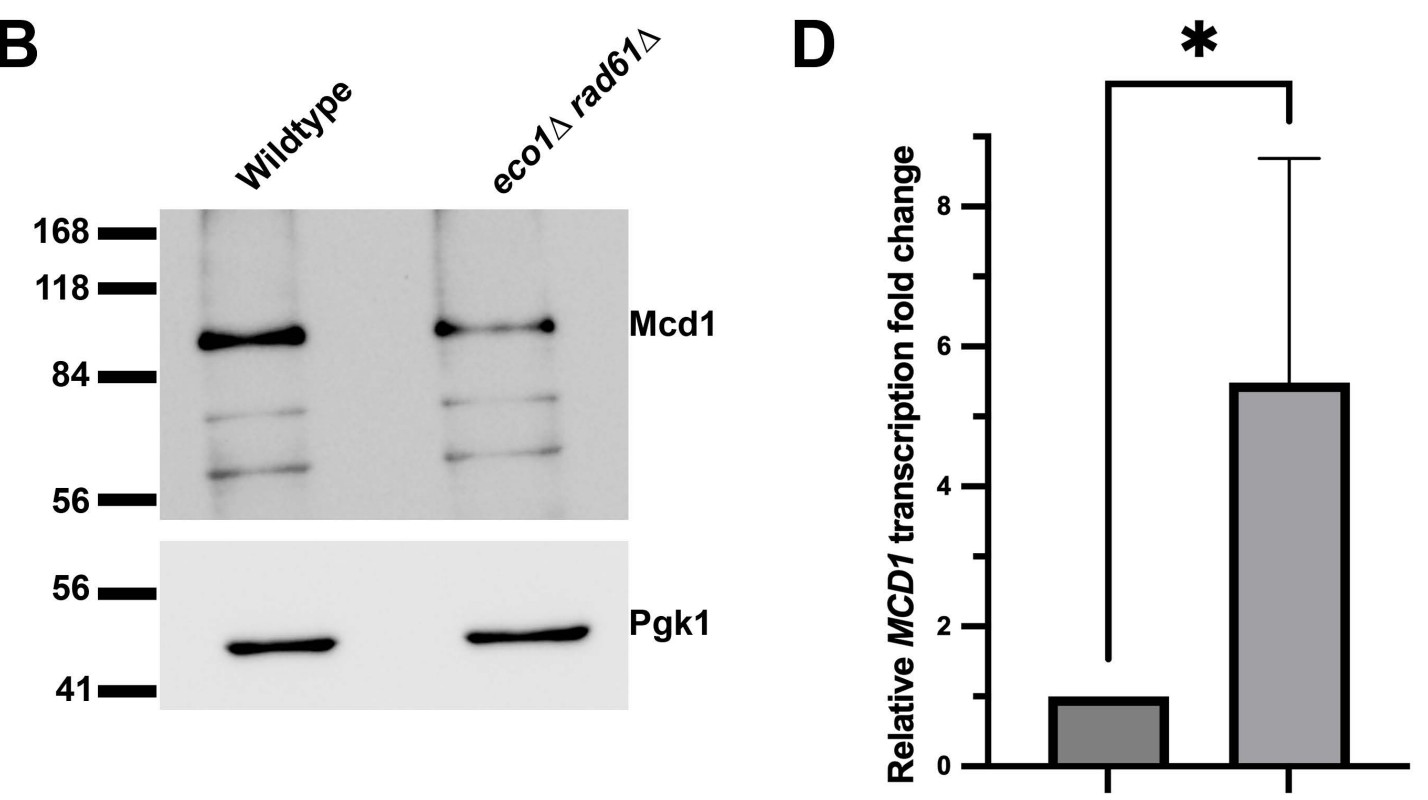

**Fig 3. *MCD1* mRNA expression is increased in *eco1Δ rad61Δ* double mutant cells. (A)** Flow cytometry data of DNA content for log phase wildtype (YPH499) and *eco1Δ rad61Δ* (YBS829) double mutant cells arrested in S phase at 30°C for 3 hrs. **(B)** Representative Western Blot of Mcd1 (top panel) and Pgk1 (lower panel) protein obtained from extracts of HU-synchronized wildtype and *eco1Δ rad61Δ* double mutant cells indicated in **(A)**. **(C)** Quantification of Mcd1, normalized to Pgk1 loading controls. Statistical analysis was performed using a two-tailed *t*-test. Statistical differences (**) are based on a $P < 0.01$ obtained across four experiments (n = 4). Error bars indicate the standard deviation. **(D)** Quantification of *MCD1* mRNA fold change normalized to the expression of the housekeeping gene *ALG9*. Statistical analysis was performed using a two-tailed *t*-test. Statistical differences (*) are based on a $P < 0.05$ obtained across four experiments (n = 4). Error bars indicate the standard deviation.

cohesin alleles (i.e., Mcd1 loss is a downstream consequence of cohesin inactivation, but otherwise unimportant) or due to the reduction in Mcd1? To differentiate between these two possibilities, we tested the extent to which elevated expression of *MCD1* could suppress the lethality of cells that harbor ts mutations in other cohesin genes. Wildtype, *eco1–1*, *scc3–6*, *smc3–42*, and *smc1–259* cells were each transformed with either vector alone or vector that drives elevated expression of *MCD1*. Log phase cultures of the resulting transformants were then serially diluted onto selective media plates and incubated across a range of temperatures. As expected, elevated *MCD1* expression had no effect on the growth of wildtype cells at the temperatures tested. In contrast, overexpression of *MCD1* suppressed the ts growth defects in all five cohesin ts alleles (Fig 7) - in some cases up to 37°C. We further found that elevated *MCD1* expression suppressed the ts growth defect of *scc2–4* mutant cells, cells that contain wildtype alleles of each cohesin gene subunit (Fig 7E). Importantly, the suppression of growth defects is unique to *MCD1*: overexpression of *ECO1* failed to rescue the ts growth defects of *smc3–42*, *smc1–259*, *mcd1–73*, and *pds5–3* cells [99] and overexpression of *PDS5* failed to rescue the ts growth defects of *eco1Δ rad61Δ* (S1D Fig). Nor did the deletion of *RAD61* (which is required for *eco1Δ* cell viability [70,73,100]), rescue the viability of *smc3–42* or *mcd1–1* mutant strains [101]. These results provide evidence that the targeted loss of Mcd1 significantly contributes to the lethality of cohesin-mutated cells, and also confound prior interpretations of the severity of phenotypes attributed solely to those ts alleles.

## Loss of Mcd1 differentially impacts cohesion and condensation

Above, we established loss of Mcd1 as a key driver of most, if not all, cohesin-mutated strain lethalities. It next became important to test the extent to which Mcd1 loss exacerbated cohesin functions. Wildtype and *smc1–259* strains were genetically modified to contain either an rDNA condensation marker (Net1-GFP) or a cohesion assay cassette (tetO and TetR-GFP) [11,12,14,76,78,102,103]. The modified strains were then transformed with a high-copy vector alone or vector that drives elevated expression of *MCD1*. Log phase cultures of the resulting transformants were arrested in G1 (alpha factor, αF) and then released into 34°C (non-permissive for *smc1–259* cells) rich medium that contains nocodazole (NZ) to arrest cells in preanaphase. DNA content (flow cytometry) and cell morphologies were monitored at various stages of both experiments (Figs 8A, 9A).

Notably, *smc1–259* mutant cells have not been previously assessed for condensation defects. Here, we exploited the well-established analysis of rDNA chromatin architecture using Net1-GFP [11,14,76,78,104]. In wildtype cells, rDNA converts from a decondensed puff-like structure during G1 into tight loops (occasionally observed as bars) during mitosis [105]. As expected, wildtype cells arrested in preanaphase exhibited well-defined rDNA loops, indicative of robust chromosome condensation (Fig 8B, 8C). In contrast, only 21% of *smc1–259* cells contained tight loops such that the majority of cells exhibited defects in rDNA condensation (Fig 8B, 8C). These results extend prior findings regarding the condensation defects exhibited by other cohesin mutated strains [11,32,48,75,101,106,107]. Elevated expression of *MCD1* had no observable impact on rDNA structure in wildtype cells. Surprisingly, *MCD1* expression produced only a modest increase (38%, compared to 21% in the vector control) in the percent of *smc1–259* cells that contained condensed rDNA loops (Fig 8C). Given prior evidence that suppression of condensation defects underlies the improved viability of cohesin-mutated cells [108], we decided to further investigate the condensation of rDNA structure. Mutated cells that did not contain tight and well-defined rDNA loops were parsed into two categories: puff-like (fully decondensed) and in which some structure

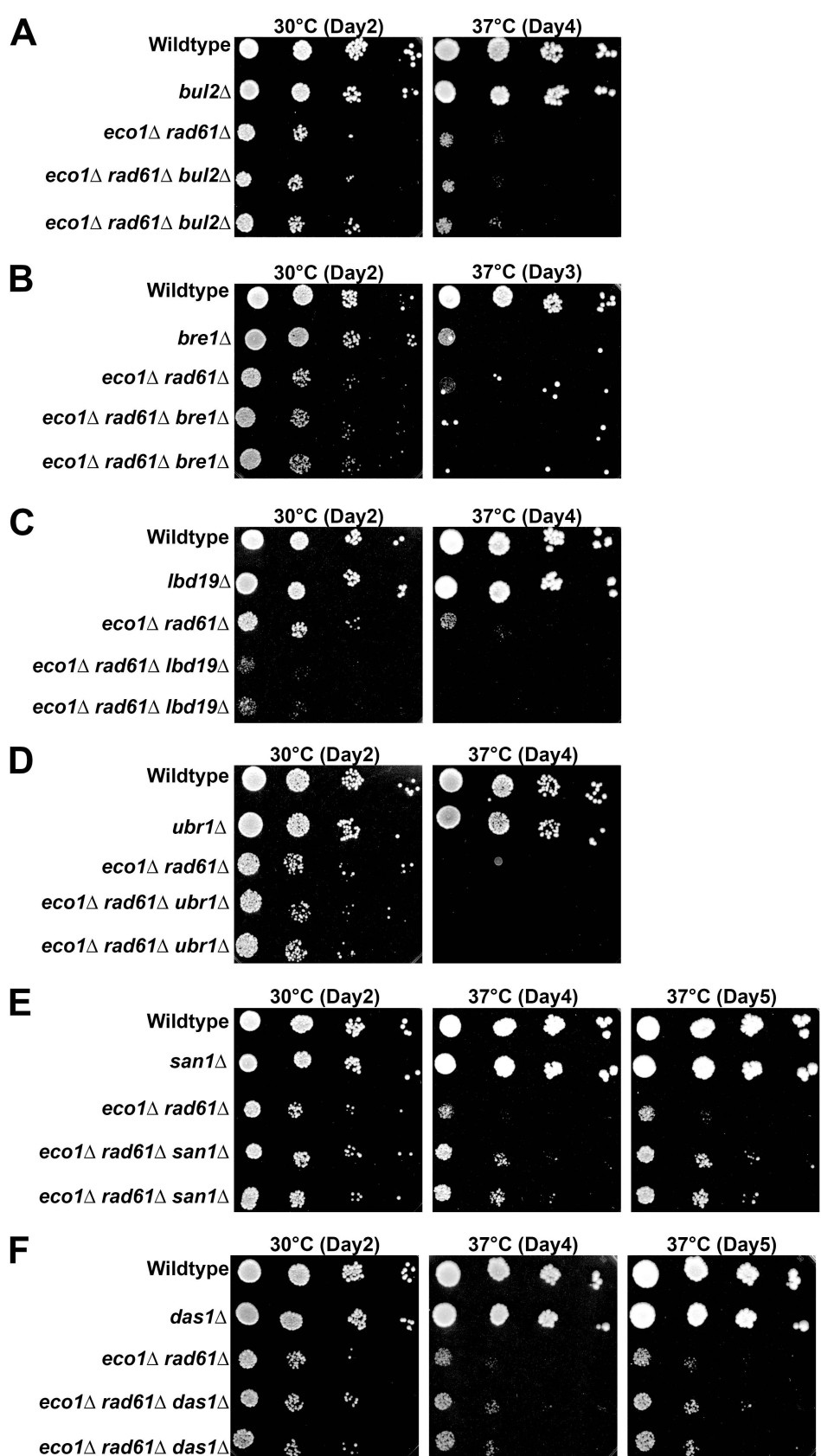

**Fig 4. Deletion of Ubiquitin E3 ligases *SAN1* and *DAS1* suppress the growth defects of *eco1Δ rad61Δ* double mutant cells. (A)** Growth of 10-fold serial dilutions of wildtype (YPH499), *bul2Δ* (YGS277), *eco1Δ rad61Δ* (YBS829) and two independent isolates of *eco1Δ rad61Δ bul2Δ* triple mutant cells (YGS279, YGS280); **(B)** wildtype (YPH499), *bre1Δ* (YGS309), *eco1Δ rad61Δ* (YBS829) and two independent isolates of *eco1Δ rad61Δ bre1Δ* triple mutant cells (YGS292, YGS293); **(C)** wildtype (YPH499), *ldb19Δ* (YGS281), *eco1Δ rad61Δ* (YBS829) and two independent isolates of *eco1Δ rad61Δ ldb19Δ* triple mutant cells (YGS282, YGS283); **(D)** wildtype (YPH499), *ubr1Δ* (YGS352), *eco1Δ rad61Δ* (YBS828) and two independent isolates of *eco1Δ rad61Δ ubr1Δ* triple mutant cells (YGS354, YGS355). **(E)** wildtype (YPH499), *san1Δ* (YGS284), *eco1Δ rad61Δ* (YBS829) and two independent isolates of *eco1Δ rad61Δ san1Δ* triple mutant cells (YGS286, YGS287); and **(F)** wildtype (YPH499), *das1Δ* (YGS288), *eco1Δ rad61Δ* (YBS829) and two independent isolates of *eco1Δ rad61Δ das1Δ* triple mutant cells (YGS290, YGS291). Temperature and days of growth are indicated.

was apparent within the rDNA mass (partial decondensation). Focusing on the more severe of the two phenotypes, *smc1–259* cells that contained vector alone were strongly biased toward the frequency of puffs (~65% puffs compared to 12% partial condensed) (Figs 8D, S4). *smc1–259* cells in which *MCD1* was over expressed, however, contained a significant decrease in the frequency of puffs (25%, down from 65% for vector alone) (Fig 8D). In combination, these findings reveal that the loss of Mcd1 significantly contributes to the condensation defects that might otherwise be attributed to the *smc1–259* allele.

Next, we assessed the effect of reduced Mcd1 on sister chromatid cohesion. In mitotic wildtype cells, the combination of GFP-tetR/tetO, which marks each sister chromatid, document close tethering such that both loci appear as a single dot. In cohesin mutated cells, separated sisters are readily detected as two dots [12]. Elevated expression of *MCD1* in wildtype cells resulted in normal frequencies of tethered sister chromatids (1 dot/nucleus), similar to vector alone (Fig 9B, 9C). *smc1–259* cells that contained vector alone exhibited a high (60%) frequency of 2 dots/nucleus (Fig 9B, 9C), consistent with prior studies and the frequency of cohesion defects observed in other cohesin mutated cells [11,48,80,101,106,108]. Notably, elevated expression of *MCD1* significantly restored sister chromatid tethering in *smc1–259* cells, with only 35% (compared to 60% vector control) of cells exhibiting cohesion defects. In combination, the above findings reveal a surprisingly biased role for Mcd1 in promoting cohesion with a more nuanced role in promoting chromosome condensation.

## Discussion

The cohesin component Mcd1, which caps the ATPase domains of Smc1 and Smc3 to form the core cohesin complex, is greatly reduced in cells that harbor mutations in nearly every cohesin gene tested to date [64–67]. A priori, a simple explanation is that cohesin gene mutations destabilize the cohesin ring and, in some unknown fashion, promote the loss of the non-mutated Mcd1 protein. The first revelation of the current study is that Mcd1 levels are also significantly reduced in cells mutated for regulatory mechanisms in which the cohesin complexes remain intact. For example, cells that are fully absent of *ECO1* and *RAD61* cells exhibit reduced Mcd1. Strikingly, Mcd1 is significantly reduced even in *rad61Δ* cells in which cohesins appear hyper-stabilized, exhibit extended DNA-associations, and are fully functional to condense chromosomes and extrude DNA loops [15,65,72,73,75–79,109]. Mcd1 is also reduced in *scc2–4* cells, which exhibit cohesin loading defects, but have no known defect in either cohesin complex assembly or integrity [80–83]. These findings, coupled with the recognition that it is wildtype Mcd1 protein that is reduced, not a mutated version, are inconsistent with a simple instability or integrity model. Intriguingly, reduction of RAD21, the human ortholog of Mcd1, appears to be a conserved feature of cells that harbor cohesin gene mutations [110–114], but the significance of either Mcd1 or RAD21 reduction remained largely unrecognized. In summary, our current study suggests that cohesin function is strongly regulated through Mcd1/RAD21 stability, and that manipulating its levels may have significant implications in cohesinopatheis and cancer research.

The second set of findings reported here impact all prior analyses of cohesin mutated strains. Previously, phenotypes exhibited by mutated cohesin strains were interpreted to reflect protein inactivation/misfolding of the particular gene under study. Instead, we find substantial suppression of the ts growth defects that occur in cohesin mutated cells (*smc1–259*, *smc3–42*, *scc3–6*, *scc2–4* and *eco1^ctf7-203*) simply by re-elevating Mcd1 levels. Moreover, our results differentiate Mcd1

**A**

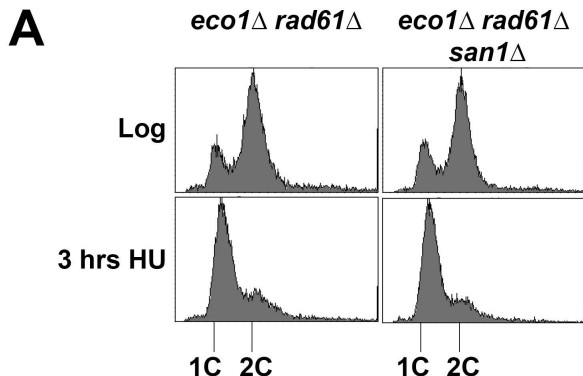

**B**

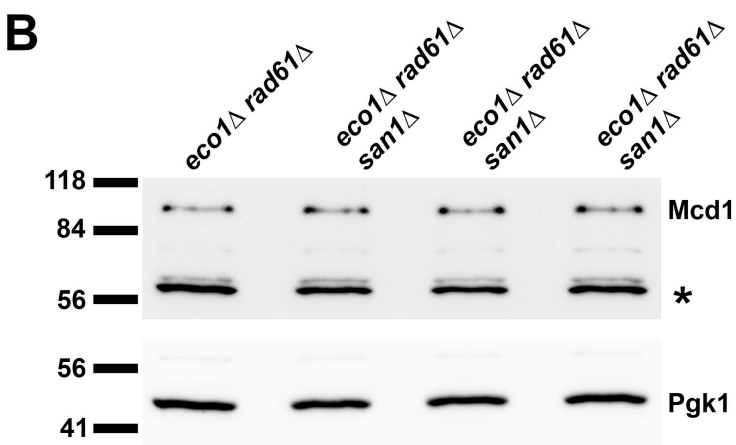

**C**

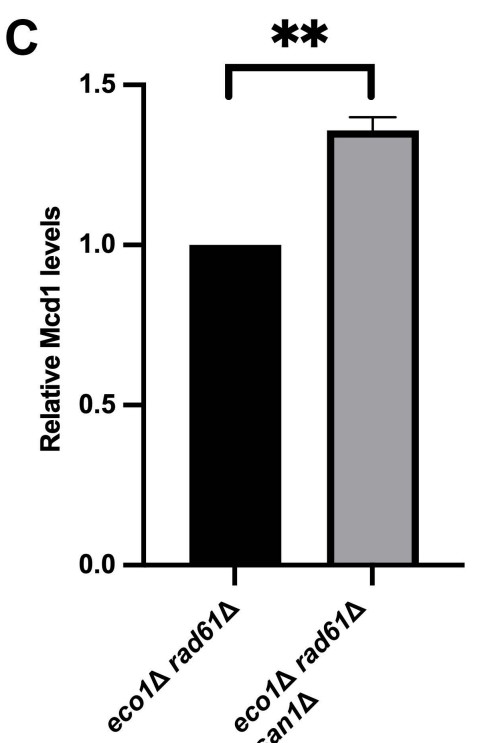

**Fig 5. San1 promotes Mcd1 degradation in the *eco1Δ rad61Δ* double mutant cells. (A)** Flow cytometry data of DNA content for log phase of *eco1Δ rad61Δ* (YBS829) double mutant cells and 3 biological replicates of *eco1Δ rad61Δ san1Δ* (YGS290) arrested in S phase at 30°C for 3 hrs. **(B)** Western Blot of Mcd1 (top panel) and Pgk1 (lower panel) protein obtained from extracts of HU-synchronized cells indicated in **(A)**. * indicates non-specific band. **(C)** Quantification of Mcd1, normalized to Pgk1 loading controls. Statistical analysis was performed using a one sample t-test, two tailed. Statistical differences (**) are based on a $P \leq 0.01$ obtained across three biological replicates (n = 3). Error bars indicate the standard deviation.

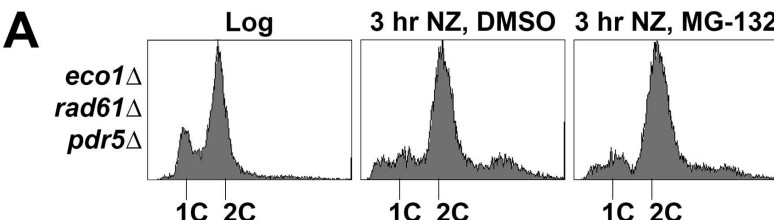

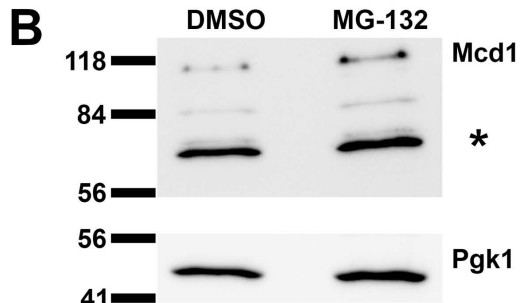

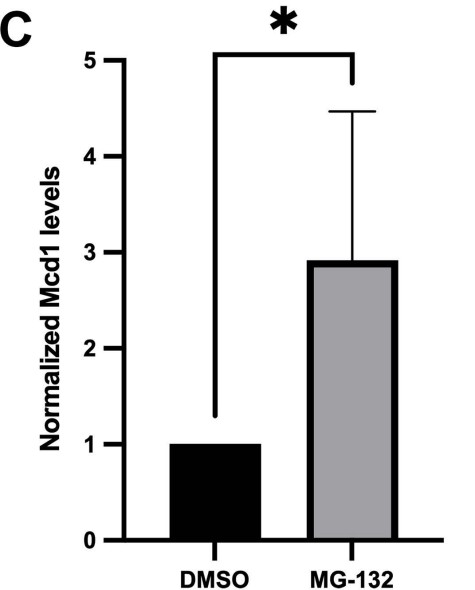

**Fig 6. Addition of MG-132 in M phase arrested cells elevates Mcd1 levels. (A)** Flow cytometry data of DNA content for log phase of *eco1Δ rad61Δ pdr5Δ* cells (YGS377) arrested in M phase with nocodazole at 30°C for 3 hours in the presence of DMSO (control) or MG-132 proteosome inhibitor. **(B)** Representative western Blot of Mcd1 (top panel) and Pgk1 (lower panel) protein obtained from extracts of M phase synchronized cells indicated in **(A)**. * indicates non-specific band. **(C)** Quantification of Mcd1, normalized to Pgk1 loading controls. Statistical analysis was performed using a two-tailed *t*-test. Statistical differences (*) are based on a $P < 0.05$ obtained across four experiments (n = 4). Error bars indicate the standard deviation.

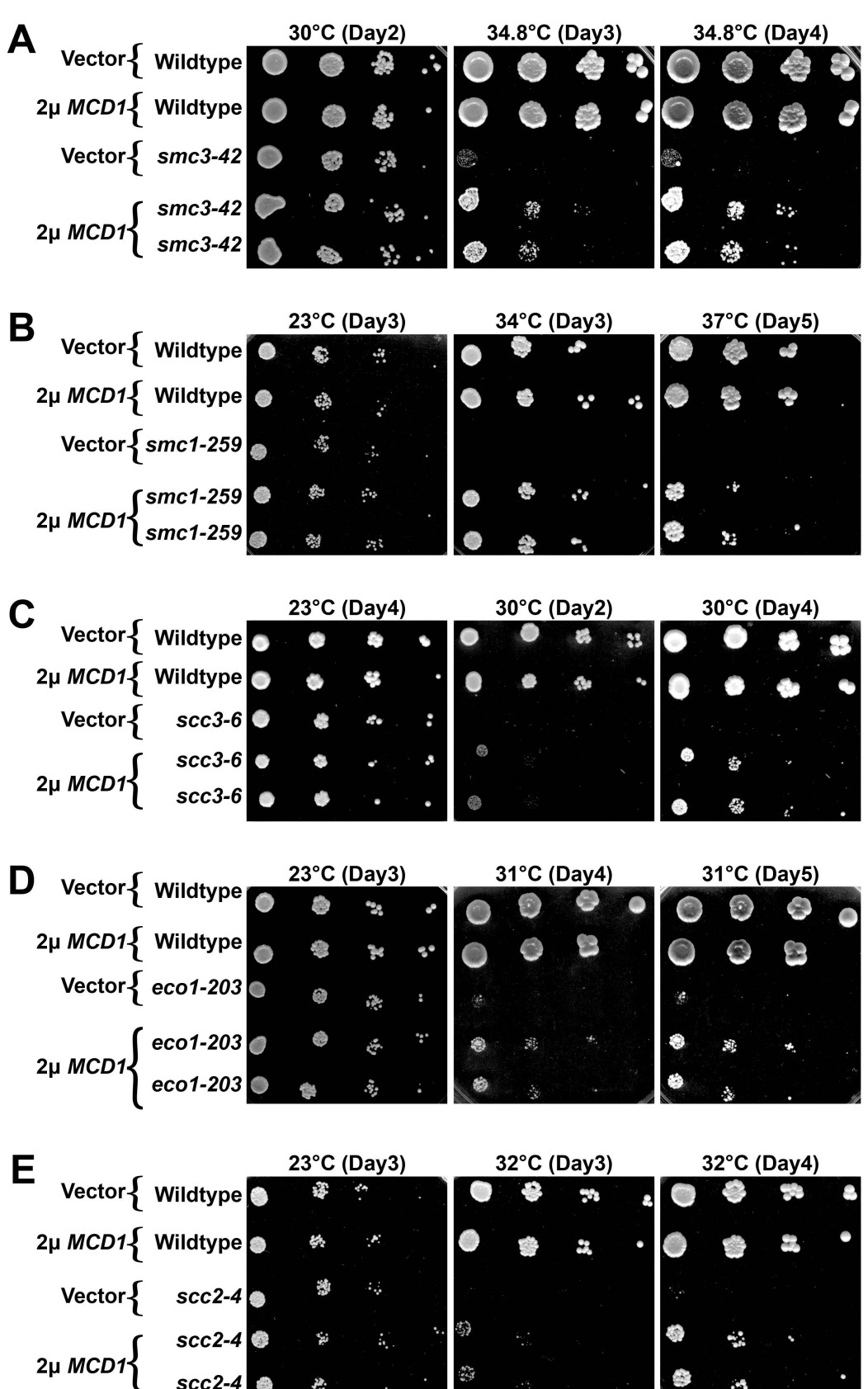

**Fig 7. Increased Mcd1 levels partially rescue the growth defects of cohesin mutated cells.** Growth of 10-fold serial dilutions of cells (strains indicated below) that contains either 2μ vector (pRS424) or 2μ vector that contains *MCD1* (pBS1476). Two independent isolates are shown of mutated strains that express elevated levels of *MCD1*. Cell strains are as follows for **(A-E)**: wildtype (YGS209, YBS4558, YGS216), *smc3-42* (YGS229), *smc1-259* (YGS211), *scc3-6* (YBS4568), *eco1<sup>ctf7</sup>-203* (YGS329), and *scc2-4* (YGS218) all contain vector alone. Wildtype (YGS210, YBS4562, YGS217), *smc3-42* (YGS230, YGS231), *smc1-259* (YGS212, YGS213), *scc3-6* (YBS4569), *eco1<sup>ctf7</sup>-203* (YGS330, YGS331), and *scc2-4* (YGS219, YGS220) all contain elevated *MCD1* expression. Temperature and days of growth are indicated. Strain genotypes are provided in S1 Table.

roles in cohesion and condensation: re-elevating Mcd1 levels produced an overtly increased suppression of cohesin defects, compared to condensation defects, in *smc1–259* cells. We infer from these findings two key principles. First, that Mcd1-dependent restoration of cohesion (and to a lesser extent condensation) primarily accounts for the decrease in temperature-sensitive growth of *smc1–259* cells. Second, that Smc1 appears to exert a more significant role in condensation (and possibly less a significant role in cohesion) than previously reported. In combination, these findings suggest that a re-evaluation of phenotypic severity, previously attributed to cohesin gene mutations, is warranted. More broadly, studies that characterize phenotypes for a mutated component within a complex should be coupled with analyses regarding the persistence of the remaining subunits.

The third set of findings that emerge from the current study is the identification of the pathway through which Mcd1 is degraded. In unperturbed cells, Mcd1 is degraded at anaphase onset by APC-dependent activation of Esp1 [84–86,115]. Here, our findings largely negate a role for an Esp1-dependent mechanism and instead document for the first time that E3 ligases (San1 and Das1) and the proteosome perform unanticipated roles in Mcd1 degradation. Importantly, Mcd1 levels are significantly decreased in both S and M phases in response to cohesin dysfunction. While the proteosome plays a key role in Mcd1 degradation during mitosis (preanaphase onset), proteosome inhibition failed to restore Mcd1 levels during S phase despite roughly a 5-fold increase in *MCD1* transcription. These findings expose a critical gap in knowledge regarding the cell cycle phase of E3 ligase activation that is required to reduce Mcd1 during S phase. Moreover, the E3 ligase responsible for Mcd1/RAD21 degradation remain largely unidentified. Future studies aimed at defining this regulatory layer will not only advance our understanding of cohesin regulation but may also provide means to modulate cohesin function through Mcd1 stability, offering new avenues for cohesin-targeted cancer therapies.

Another revelation of the current study relates to the mechanism through which yeast cells appear to achieve homeostatic levels of Mcd1. Subsequent to degradation at anaphase onset, cells initiate a new round of *MCD1* transcription at the G1/S transition that results in Mcd1 protein levels that peak during S phase [11]. Our results suggest that Mcd1 protein may negatively regulate its own expression such that E3-ligase degradation of Mcd1 (in response to cohesin deficiency) results in a dramatic upregulation in *MCD1* expression during S phase that is well above that of wildtype cells. Given reduced Mcd1 levels exhibited by cohesin-defective cells, degradation appears to be heavily favored over the increase (~5-fold) in transcription, suggesting that the E3 ligases that promote Mcd1 degradation are likely significantly upregulated.

The identification of surveillance mechanisms that target Mcd1/RAD21 may significantly impact studies of both human development and cancer progression. However, a limitation of the current study is the lack of a molecular description of how cells detect cohesin dysfunction and, in response, activate Mcd1 degradation. Of several possibilities, cohesion defects that result in the retention of an activated spindle assembly checkpoint may promote the targeting of Mcd1 in mitotic cells or cells that progress into G1. Another possibility is that cohesin-dependent transcriptional defects impact the expression of any number of genes, for example those that encode for Mcd1, E3 ligase inhibitors, or factors that promote cohesin function. More recent evidence suggests that cyclin-dependent kinases (CDKs) also play a critical role in Mcd1 regulation. For instance, deletion of the G1 cyclin, *CLN2*, rescues both *eco1Δ rad61Δ* cell ts growth defects and *pds5Δ elg1Δ* lethality - in both cases by increasing Mcd1 levels [64,77]. These findings raise additional possibilities, including that 1) Cln2-CDK modifies cohesin directly in order to recruit E3 ligases to Mcd1, 2) Cln2-CDK targets/activates E3 ligases (such as San1) to degrade Mcd1, or 3) cohesin represses in some manner the transcription of *CLN2* that otherwise acts

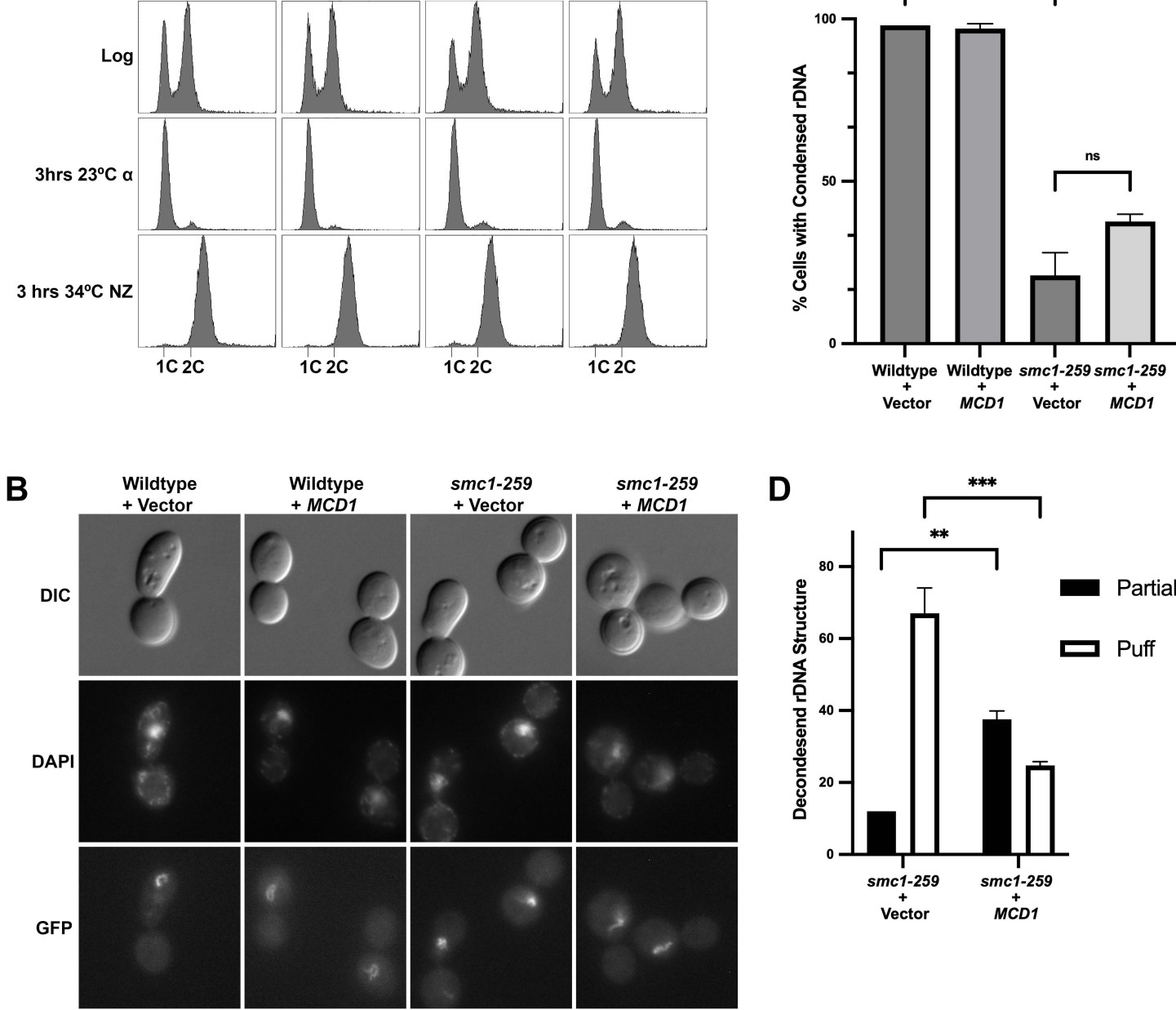

**Fig 8. Increased Mcd1 protein levels suppress *smc1-259* cell condensation defects.** (A) Flow cytometry data of DNA content for log phase cells pre-synchronized in G1 phase at 23°C, then shifted to 34°C (the non-permissive temperature of *smc1-259*) in nocodazole. Genotypes of wildtype (YGS335, YGS337) and *smc1-259* mutated (YGS338, YGS341) cells that contain either 2μ vector (pRS424) or 2μ vector that contains *MCD1* are provided in S1 Table. (B) Representative micrographs of rDNA detected by Net1-GFP. DNA is detected by DAPI staining. (C) The percentage of cells with condensed rDNA is plotted. At least 120 nuclei were scored per genotype. Statistical analysis was performed using a two-tailed *t*-test. Statistical differences (**) are based on a *P* < 0.01 obtained across two experiments (n=2). ns indicates not significantly different. Error bars indicate the standard deviation. (D) The uncondensed rDNA structures for all strains were further classified as either fully decondensed "puffs" or partially decondensed "partial". Statistical analysis was performed using a two-tailed *t*-test. Statistical differences (**) are based on a *P* < 0.01, and (***) are based on a *P* < 0.001 obtained across two experiments (n=2). Error bars indicate the standard deviation.

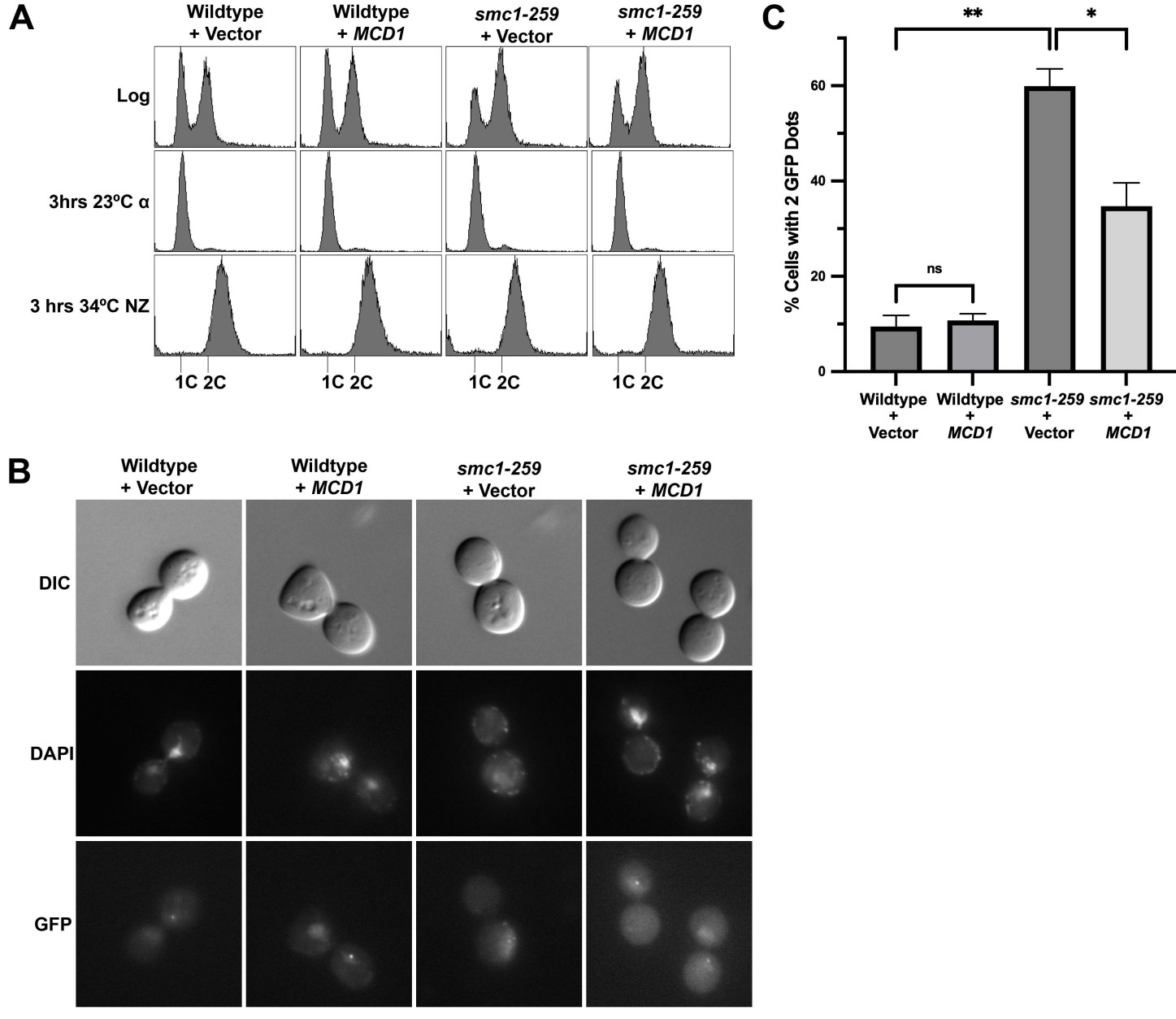

**Fig 9. Increased Mcd1 levels significantly suppress sister chromatid cohesion defects in *smc1-259* cells.** (A) Flow cytometry data of DNA content as described in Fig 8 (B) Representative micrographs of GFP dots (markers of sister chromatid cohesion) in cell treatments as described in Fig 8. Genotypes of wildtype (YGS333, YGS334) and *smc1-259* mutated (YGS321, YGS323) cells modified to contain both cohesion cassettes and either 2μ vector (pRS424) or 2μ vector that contains *MCD1* are provided in S1 Table. (C) The percentage of cells in which sisters are separated (two GFP spots indicated of a sister chromatid cohesion defect) is plotted. At least 120 cells were scored for each genotype. Statistical analysis was performed using a two-tailed *t*-test. Statistical differences (ns) are based on a P > 0.05, (*) are based on a *P* < 0.05 and (**) are based on a *P* < 0.01 obtained across two experiments (n=2). Error bars indicate the standard deviation.

to limit *MCD1* transcription. Exploring the pathways through which signals, that arise from cohesin defects, are relayed to E3 ligases constitutes the next important advancement in our understanding of cohesin biology.

Finally, Mcd1 targeting appears to extend beyond cells that exhibit cohesin defects. For instance, exposing cells to reactive oxygen species also triggers Mcd1 degradation via an apoptotic pathway [87,116–118]. The conserved nature of a surveillance mechanism that targets Mcd1 is supported by findings that RAD21 (homolog of Mcd1) is degraded by caspases 3 and 7 during apoptotic responses [87,119,120]. Together, these results suggest that inactivating cohesin functions through Mcd1/RAD21 degradation (through E3 ligases or caspases) may represent an evolutionarily conserved mechanism for promoting cell death. Conversely, cells that evade this surveillance system may obtain proliferative benefits. In support of this model, elevated expression of almost every cohesin subunit (RAD21, SMC1A, SMC3, STAG1, PDS5, WAPL) and cohesin regulator (NIPBL, MAU2) appears to be oncogenic and present in a wide range of cancer cells [44,45,47]. In fact, 30 of 33 different cancer types were recently shown to express elevated levels of ESCO2 [46]. RAD21, however, is the most frequently amplified cohesin in non-small-cell lung cancer, breast cancer cells, cervical cancer, ovarian cancer, oral cancer and prostate cancer [40,121–128]. How cancer cells circumvent the transcriptional and degradational mechanisms that normally regulate RAD21 levels remains unclear, but reducing RAD21 levels in cancer cells appears effective in reducing cell proliferation and/or inducing apoptosis [121,122,127,128]. These findings, in combination with our current study, highlight the potential for targeting Mcd1, the master regulator of cohesin function, as a novel therapeutic strategy across multiple cancer types.

## Materials and methods

### Yeast strains, media, and growth conditions

All strains (see S1 Table for strain genotypes) were grown on YPD-rich media unless placed on selective medium to facilitate plasmid transformation/retention or spore identification [129]. Cells that required inhibition of the proteasome, were treated with 75μM MG-132 (MedChemExpress Cat No. HY-13259) and a control of DMSO.

### Strain generation

Primers used to delete genes (*BUL2*, *BRE1*, *LDB19*, *DAS1* and *SAN1*) and verify proper integration are listed in S2 Table. GFP-tagging Net1, to include either *kanMX6* or *TRP1* markers, are previously described [130]. The cohesion assay strains used in this study (YGS333, YGS334, YGS321, YGS323) were generated by crossing *smc1–259* (YBS3168/K6013) with wildtype cells that harbor the cohesion assay cassette (*tetO:URA3 tetR-GFP:LEU2 Pds1-Myc:TRP1*) (YBS1042) [103,131]. The resulting diploid (YGS301) was sporulated and dissected to obtain *smc1–259 tetO:URA3 tetR-GFP:LEU2* cells and wildtype cassette strain *tetO:URA3 tetR-GFP:LEU2*.

Wildtype and cohesin mutant cells overexpressing vector were transformed with either pRS424 plasmid (2μ *TRP1*) or pRS425 (2μ *LEU*) and cells overexpressing *MCD1* were transformed either with pRS425 (2μ *LEU*) or pGS35 (2μ *LEU2 MCD1*) [65]. See S1 Table for resulting strain names and genotypes.

### Western blots

Cell numbers for each log phase strain were normalized to 2 $OD_{600}$. Whole cell protein extracts were prepared as described in [132] with minor modifications. Cells were mechanically lysed (Bead-beater, BioSpec) in 17% TCA with regular intermittent cooling on ice. The beads were washed two times in 500μL of 5% TCA and the two lysates combined and centrifuged at 15,000 rpm for 20 min at 4°C with subsequent solubilization in 3% SDS and 0.25 M Tris-base buffer. Western blotting and protein detection using the anti-Mcd1 and anti-Pds5 antibodies (generous gift from Dr. Vincent Guacci and the lab of Doug Koshland), anti-PGK1 (Invitrogen), Goat anti-Mouse HRP (BIO-RAD) or Goat anti-Rabbit HRP (BIO-RAD), were performed as previously described [65]. Protein band intensities (obtained by ChemiDoc MP) were quantified using Image J. Significance was determined by a two-tailed test as described in legends.

## RNA extraction and qRT-PCR

Cell numbers from log phase cultures were normalized to 2 $OD_{600}$, pelleted by centrifugation and frozen in liquid nitrogen. Cells were lysed mechanically using a bead-beater (BioSpec) for 8 min with intermittent cooling on ice. RNA was extracted and purified using the RNeasy Mini Kit (Qiagen) per manufacturer's instructions and quantified using a nanodrop (Thermo Scientific, NanoDrop One$^C$). Normalized RNAs were treated with Turbo DNase (Ambion) and then reverse transcribed using SupercriptIII (Invitrogen). Quantitative Real -Time (qRT) PCR was performed in triplicates using the Rotor-Gene SYBR Green PCR kit (Cat. No. 204074) and $C_T$ values measured using the Rotor Gene (Corbett). $C_T$ values of *MCD1* and internal control ALG9 were averaged and the fold change in *MCD1* expression determined using the $2^{-\Delta\Delta Ct}$ method [133].

## Condensation and cohesion assays

Cohesion and condensation assays were performed as previously described [102] with the following modifications. Log phase cells were grown in selective media, followed by pre-synchronized in G1 (alpha factor) for 3 hr at 23°C in YPD- rich media. The resulting cultures were harvested, washed 2 times and then shifted to 37°C for 3 hr in fresh media supplemented with nocodazole. Cell aliquots of the resulting preanaphase arrested cells were fixed at room temperature in paraformaldehyde to a final concentration of 3.7%. Cells were assayed using an E800 light microscope (Nikon) equipped with a cooled CD camera (Coolsnapfx, Photometrics) and imaging software (IPLab, Scanalytics).

## Flow cytometry and cell cycle progression

Log phase cultures were normalized ($OD_{600}$) and synchronized at specific cell stages using the following treatments: early S phase with 0.2 M Hydroxyurea (SIGMA, H8627), G1 phase with 3 µM alpha factor (ZYMO RESEARCH, Y1001), M phase with 20 µg/ml of nocodazole (SIGMA, M1404). Log phase growth and proper cell cycle arrest were confirmed by flow cytometry as previously described [102,109].

## Supporting information

**S1 Fig. Pds5 protein levels are maintained in the *eco1Δ rad61Δ* mutated cells, and *PDS5* overexpression does not rescue *eco1Δ rad61Δ* cells growth defects.** (A) Flow cytometry data of DNA content for log phase wildtype (YPH499) and *eco1Δ rad61Δ* (YBS828) double mutant cells arrested in S phase at 30°C for 3 hrs. (B) Representative Western Blot of Mcd1 (top panel) and Pds5 (middle) and Pgk1 (lower panel) protein obtained from extracts of HU-synchronized wildtype and *eco1Δ rad61Δ* mutant cells indicated in (A). * indicates non-specific band. (C) Quantification of Pds5, normalized to Pgk1 loading controls. Statistical analysis was performed using a two-tailed *t*-test. Statistical differences (ns) are based on a $P > 0.05$ obtained across three experiments (n = 3). Error bars indicate the standard deviation. (D) Growth of 10-fold serial dilutions of wildtype cells overexpressing vector alone (YBS4067) or overexpressing *PDS5* (YBS4071) and *eco1Δ rad61Δ* overexpressing vector alone (YBS4069) or overexpressing *PDS5* (YBS4075, YBS4076).
(TIF)

**S2 Fig. Mutation of *ESP1* fails to suppress the growth defects of *eco1Δ rad61Δ* double mutant cells.** (A) Schematic key for the following strains: wildtype (YPH499), *eco1Δ rad61Δ esp1–1* (YBS4864*), esp1–1* (YBS4862, YBS4863), *eco1Δ rad61Δ* (YBS4860, YBS4861). (B) Streak assay for the strains indicated in (A) at 23°C and 34°C.
(TIF)

**S3 Fig. Addition of MG-132 in S phase arrested cells fails to increase Mcd1 levels.** (A) Flow cytometry data of DNA content for log phase of *eco1Δ rad61Δ pdr5Δ* arrested in S phase at 30°C for 2 hours, then treated with DMSO (control) or MG-132 in the continued presence of HU for an additional 2 hours. (B) Western Blot of Mcd1 (top panel) and Pgk1 (lower panel) protein obtained from extracts of HU-synchronized cells in DMSO or MG132 indicated in (A). "A" and "B" indicate independent biological replicates (YGS375, YGS377). * indicates non-specific band. (C) Quantification of Mcd1,

normalized to Pgk1 loading controls. Statistical analysis was performed using a two-tailed $t$-test. Statistical differences (ns) are based on a $P > 0.05$ obtained across two biological replicates (n = 2). Error bars indicate the standard deviation. (TIF)

**S4 Fig. Zoomed-in micrographs showing the effects of *MCD1* overexpression on sister chromatid cohesion and rDNA condensation in *smc1–259* cells.** (A) Enlarged micrographs from Fig 8B. Top panel: wildtype cells overexpressing vector alone. Middle panel: *smc1–259* cells overexpressing vector alone. Bottom panel: *smc1–259* cells overexpressing *MCD1*. Yellow arrow indicates a condensed, puff, or partially condensed rDNA (see Fig 8 legend). (B) Enlarged micrographs from Fig 9B. Top panel: wildtype cells overexpressing vector alone. Middle panel: *smc1–259* cells overexpressing vector alone. Bottom panel: *smc1–259* cells overexpressing *MCD1* (see Fig 9 legend). (TIF)

**S1 Table. Strains used in this study.**
(PDF)

**S2 Table. Oligonucleotides used in this study.**
(PDF)

## Acknowledgments

The authors gratefully acknowledge the generous gifts of Mcd1-directed and Pds5-directed antibodies, as well as numerous yeast strains from Dr. Vincent Guacci and the Koshland Lab. The authors also thank Skibbens lab members and participants in the Super Group Yeast Club for helpful discussions during the preparation of this paper.

## Author contributions

**Conceptualization:** Gurvir Singh, Robert V. Skibbens.

**Data curation:** Gurvir Singh.

**Formal analysis:** Gurvir Singh.

**Investigation:** Gurvir Singh.

**Methodology:** Gurvir Singh.

**Project administration:** Gurvir Singh, Robert V. Skibbens.

**Resources:** Robert V. Skibbens.

**Supervision:** Robert V. Skibbens.

**Validation:** Gurvir Singh.

**Visualization:** Gurvir Singh.

**Writing – original draft:** Gurvir Singh.

**Writing – review & editing:** Gurvir Singh, Robert V. Skibbens.

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
