## [Decision Letter · Decision Letter 0]

12 Sep 2025

PGENETICS-D-25-00868

Aberrant cohesin function in Saccharomyces cerevisiae

activates Mcd1 degradation to promote cell lethality

PLOS Genetics

Dear Dr. Skibbens,

Thank you for submitting your manuscript to PLOS Genetics. As you can see the reviewers find the work of interest, but requested major revisions to the manuscript before it is suitable for publication in PLOS Genetics. Therefore, we invite you to submit a revised version of the manuscript that addresses the points raised during the review process. 

Please submit your revised manuscript within 60 days Nov 11 2025 11:59PM. If you will need more time than this to complete your revisions, please reply to this message or contact the journal office at plosgenetics@plos.org. Please include the following items when submitting your revised manuscript:

We look forward to receiving your revised manuscript.

Kind regards,

Folkert van Werven

Academic Editor

PLOS Genetics

Geraldine Butler

Section Editor

PLOS Genetics

Aimée Dudley

Editor-in-Chief

PLOS Genetics

Anne Goriely

Editor-in-Chief

PLOS Genetics

**Journal Requirements:**

At this stage, the following Authors/Authors require contributions: Robert V. Skibbens. Please ensure that the full contributions of each author are acknowledged in the "Add/Edit/Remove Authors" section of our submission form.

The list of CRediT author contributions may be found here: https://journals.plos.org/plosgenetics/s/authorship#loc-author-contributions

3) We noticed that you used the phrase 'data not shown' in the manuscript. We do not allow these references, as the PLOS data access policy requires that all data be either published with the manuscript or made available in a publicly accessible database. Please amend the supplementary material to include the referenced data or remove the references.

4) We do not publish any copyright or trademark symbols that usually accompany proprietary names, eg ©,  ®, or TM  (e.g. next to drug or reagent names). Therefore please remove all instances of trademark/copyright symbols throughout the text, including:

- ® on page: 19.

- TM on page: 16.

5) Please upload all main figures as separate Figure files in .tif or .eps format. For more information about how to convert and format your figure files please see our guidelines: 

6) We notice that your supplementary Tables are included in the manuscript file. Please remove them and upload them with the file type 'Supporting Information'. Please ensure that each Supporting Information file has a legend listed in the manuscript after the references list.

7) Please ensure that the funders and grant numbers match between the Financial Disclosure field and the Funding Information tab in your submission form. Note that the funders must be provided in the same order in both places as well.

**Reviewers' comments:**

Reviewer's Responses to Questions

**Comments to the Authors:**

Reviewer #1: In this study Singh et al propose that mcd1 protein is downregulated by E3 ligase mediated targeting in cohesin mutant yeast cells. To demonstrate this, they delete various E3 ligases in eco1Δ rad61Δ and show that deletion of two ligases reduces the growth defect observed in eco1Δ rad61Δ. In cells that contain mutation for other cohesin genes (eco1-1, scc3-6, smc3-42, and smc1-259 ) they overexpress mcd1to rescue the cohesin mutation associated defects.

The data regarding E3 ligase mediated targeting is not convincing.

Major points

1. Were the protein/RNA levels of other cohesin subunit/genes examined in eco1Δ rad61Δ? It will be good to show the levels of other cohesin subunits to determine whether only mcd1 is altered.

2. To show mcd1 targeting by E3 ligases, mcd1 protein/RNA need to be examined in in eco1Δ rad61Δ and other cohesin mutant cells in which ligases have been deleted.

3. It is unclear as to whether the authors are suggesting that E3 ligase activity is altered in cohesin defective state.

4. For the mcd1 overexpression experiments, it will be good to know what happens in eco1Δ rad61Δ cells. It will also be good to show mcd1 levels before and after overexpression of mcd1 in the wild type and cohesin mutant cells tested.

5. Mcd1 downregulation cannot be the primary mechanism for the defects seen in cells with cohesin gene mutation. It is unclear whether the authors are suggesting that mcd1 protein regulation is the primary mechanism or adds to the defects. This needs to be clarified in the manuscript. In cells with defects in core cohesin subunit like smc1-259 , the authors need to clarify how mcd1 overexpression partially restores cohesion. Even if mcd1 is increased there is still reduced smc1 and hence reduced components to form the core ring complex.

6. The impact of reduced Esp1 has been stated as data not shown. It will be good to show this data either in the main fig or supplementary data.

Minor point.

Figure 5B and 6B it would be better to have more zoomed in images. In figure 5B examples of partial and puff should be provided.

Reviewer #2: Mcd1 is an essential subunit of cohesin that is regulated throughout the cell cycle. Several studies, including previous work from this lab, have shown that Mcd1 is degraded when cohesin complex integrity is compromised. In this paper, Singh and Skibbens explore Mcd1 stability in eco1Δ wpl1Δ cells, in which chromatin-bound cohesin levels are increased, and find that under these conditions, Mcd1 levels still decrease. This finding led the authors to suggest that an active mechanism regulates Mcd1 levels. They identified ubiquitin-related factors that mediate Mcd1 degradation. In addition, they showed that reduced Mcd1 levels in Smc1 ts mutant impair cohesion but not chromosome condensation.

Cohesin is a key factor in chromatin organization, and this work suggests an interesting new paradigm for its regulation. Strengthening the mechanistic evidence would enhance the study's impact.

Major Issues

1. The regulation of Mcd1 by San1 and Das1 is shown genetically. The identification of physical interaction between these factors is based on proteomics, as described in Litwin et al., NAR, 2023. However, the confidence in the results is low, and these specific interactions have not been validated. Molecular evidence is required to support the interplay between these factors and cohesin.

2. The results show that Mcd1 overexpression in the smc1-259 background restores the cohesion defect. To better understand the mechanism, a ChIP assay should be performed to evaluate the impact on the chromatin-bound cohesin levels.

3. The first part of the study uses eco1Δ wpl1Δ cells, whereas the cell biology experiments were performed in smc1-259 cells. While Eco1 Wpl61 are regulators that are not part of the cohesin core, Smc1 is a core subunit. It hasn’t been established that cohesin complex integrity is uncompromised in smc1-259 cells and thus, that the phenotypes are fully dependent on Mcd1 degradation and not in the stability of the holocomplex. This raises the question of whether these strains share the same mechanism. The integrity of the cohesin with the smc1-259 allele in restrictive conditions should be tested.

4. Guacci et al. reported condensation defects in cells carrying the smc3-RR allele. Does Mcd1 overexpression suppress this allele as well?

Minor Points

1. Line 148: “data not shown” - all data, including negative results, should be included. Provide these results in the supplementary material.

2. Lines 216–217: “Surprisingly, MCD1 expression produced only a modest increase” - state whether this modest increase is statistically significant.

Reviewer #3: The manuscript by Singh and Skibbens provides novel mechanistic insight into how cells monitor cohesin dysfunction by showing that wild-type Mcd1 is actively degraded through a surveillance pathway involving San1 and Das1 E3 ubiquitin ligases, revealing a previously unappreciated genome integrity safeguard with implications for cohesinopathies and cancer. The work is experimentally rigorous, combining genetics, biochemistry, and imaging to directly test hypotheses, rule out alternatives, and demonstrate rescue of cohesin mutant phenotypes through Mcd1 overexpression. The data strongly support the conclusion that active degradation of Mcd1, rather than passive destabilization, drives cohesin mutant lethality, though clarifying San1/Das1’s direct role in proteasomal targeting would further strengthen the case. The manuscript is well-written, logically organized, and supported by robust quantification, with only minor typographical issues. By linking cohesin quality control to broader genome stability and disease contexts, this study offers significant and original contributions that make it a good candidate for publication in PLOS Genetics. However, this reviewer has some major concerns which needs to be addressed.

Major Comments:

(a) Direct evidence of Mcd1 proteolysis: To firmly establish that Mcd1 is actively degraded (and not just diluted or mislocalized), the authors should consider adding a direct assay for Mcd1 stability. For example, treating cells with a proteasome inhibitor during the S-phase arrest could show whether Mcd1 levels are rescued when proteolysis is blocked. Alternatively, examining ubiquitination of Mcd1 (via immunoprecipitation of Mcd1 and blotting for ubiquitin) in wild-type vs. cohesin-mutant cells would be very informative. Currently, the conclusion of proteasome-mediated degradation rests on genetic evidence (involvement of E3 ligases), which is strong but could be complemented by in vitro biochemical validation including the E2 enzyme.

(b) Mcd1 levels in suppressor mutants: As noted above, a crucial control would be to measure Mcd1 protein in the eco1Δ rad61Δ san1Δ and eco1Δ rad61Δ das1Δ strains. The manuscript shows that deleting these ligases improves growth, implying Mcd1 is preserved; actually demonstrating Mcd1 restoration in these backgrounds (via Western blot) would directly tie the suppression to Mcd1 stability.

(c) Redundancy of San1 and Das1: Since San1 and Das1 each partially alleviate the phenotype, the authors might explore whether deleting both together has an additive effect (i.e., a san1Δ das1Δ double in the eco1Δ rad61Δ background). This could establish whether these ligases act in the same pathway or have complementary roles in targeting Mcd1. If constructing that strain is not feasible, the authors should at least discuss the possibility of redundant or sequential action by these E3s. San1 typically targets misfolded proteins for degradation; therefore, the observed reduction of Mcd1 may reflect preferential elimination of its misfolded form, a distinction that should be explicitly delineated.

(d) Upstream signals: The discussion mentions Cln2-CDK as a potential upstream signal for this pathway. If possible, testing a cln2Δ mutant for suppression of Mcd1 loss (or cohesin mutant growth defects) would directly support that idea. However, this may be beyond the scope of the current work; at minimum, the authors should clarify that the Cln2 connection, while intriguing, remains speculative in this study.

(e) Rad21 amplification and elevated protein levels are observed in a significant subset of human tumors, particularly epithelial cancers. A discussion of how cancer cells override the normal proteolytic mechanisms that regulate Rad21/Mcd1 stability would help underscore the broader implications of the current findings.

I consider (a) and (b) to be important controls to include to solidify the central claim of proteolytic Mcd1 turnover.

Minor:

1. Ensure all acronyms (e.g., “ts” for temperature-sensitive) are defined on first use for clarity.

2. Use a consistent format for yeast gene and protein names. For example, RAD61 (gene) vs. Rad61 (protein) and the deletion allele (rad61Δ) should be clearly distinguished. In the manuscript, “rad61D” appears in place of “rad61Δ” – this should be corrected to the standard delta symbol to avoid confusion.

3. There are minor typos in some captions. In Figure 3, the caption begins with a duplicated panel label (“A) ... A) wildtype”), which should be corrected. Double-check all figure labels and references in the text to ensure they match (e.g., if the text refers to Figure 2A, that it indeed corresponds correctly).

4. There is a duplicated reference number in the text (ref. 95 appears twice in a row, page 12, line 255), which likely needs fixing in the bibliography. Ensure all references cited (especially in the introduction/discussion) are included and up-to-date. For example, references supporting the role of Wapl/Rad61 and Eco1 are cited; adding a brief citation or footnote in the introduction for San1 and Das1 (to note their known functions) might be helpful for readers.

5. There are some minor typos (for example, eco1D rad61D is sometimes written without the Δ symbol in the Results, and pds5Δ elg1Δ appears as pds5D elg1D in one instance).

**Have all data underlying the figures and results presented in the manuscript been provided?**

Reviewer #1: None

Reviewer #2: Yes

Reviewer #3: Yes

PLOS authors have the option to publish the peer review history of their article (what does this mean? ). If published, this will include your full peer review and any attached files.

**Do you want your identity to be public for this peer review?** For information about this choice, including consent withdrawal, please see our Privacy Policy .

Reviewer #1: No

Reviewer #2: No

Reviewer #3: No

**Figure resubmission:**
---

## [Decision Letter · Decision Letter 1]

3 Dec 2025

Dear Dr Skibbens,

We are pleased to inform you that your manuscript entitled "Aberrant cohesin function in Saccharomyces cerevisiae

activates Mcd1 degradation to promote cell lethality" has been editorially accepted for publication in PLOS Genetics. Congratulations!

Yours sincerely,

Geraldine Butler

Section Editor

PLOS Genetics

Geraldine Butler

Section Editor

PLOS Genetics

Aimée Dudley

Editor-in-Chief

PLOS Genetics

Anne Goriely

Editor-in-Chief

PLOS Genetics

BlueSky: @plos.bsky.social

Comments from the reviewers (if applicable):

Reviewer's Responses to Questions

**Comments to the Authors:**

Reviewer #2: The authors have addressed my concerns, and I recommend the study for publication.

Reviewer #3: The authors have addressed the majority of the reviewers’ comments and have undertaken substantial revisions. They have made significant efforts to improve the overall clarity, rigor, and completeness of the manuscript. The revised version reflects these improvements and is now, in my view, suitable for publication in PLOS Genetics. The authors have satisfactorily responded to my earlier concerns, providing new data where necessary, and I am satisfied with their revisions.

**Have all data underlying the figures and results presented in the manuscript been provided?**

Reviewer #2: Yes

Reviewer #3: Yes

PLOS authors have the option to publish the peer review history of their article (what does this mean? ). If published, this will include your full peer review and any attached files.

**Do you want your identity to be public for this peer review?** For information about this choice, including consent withdrawal, please see our Privacy Policy .

Reviewer #2: No

Reviewer #3: **Yes: ** Professor Debananda Pati

**Data Deposition**

http://datadryad.org/submit?journalID=pgenetics&manu=PGENETICS-D-25-00868R1

**Press Queries**

---

## [Editor Report · Acceptance letter]

PGENETICS-D-25-00868R1

Aberrant cohesin function in Saccharomyces cerevisiae

activates Mcd1 degradation to promote cell lethality

Dear Dr Skibbens,

We are pleased to inform you that your manuscript entitled "Aberrant cohesin function in Saccharomyces cerevisiae

activates Mcd1 degradation to promote cell lethality" has been formally accepted for publication in PLOS Genetics! Your manuscript is now with our production department and you will be notified of the publication date in due course.

With kind regards,

Zsofia Freund

PLOS Genetics

On behalf of:
